# Hybrid PET/MRI in Cerebral Glioma: Current Status and Perspectives

**DOI:** 10.3390/cancers15143577

**Published:** 2023-07-12

**Authors:** Karl-Josef Langen, Norbert Galldiks, Jörg Mauler, Martin Kocher, Christian Peter Filß, Gabriele Stoffels, Cláudia Régio Brambilla, Carina Stegmayr, Antje Willuweit, Wieland Alexander Worthoff, Nadim Jon Shah, Christoph Lerche, Felix Manuel Mottaghy, Philipp Lohmann

**Affiliations:** 1Institute of Neuroscience and Medicine (INM-3, INM-4, INM-11), Forschungszentrum Juelich, 52425 Juelich, Germany; n.galldiks@fz-juelich.de (N.G.); j.mauler@fz-juelich.de (J.M.); c.filss@fz-juelich.de (C.P.F.); g.stoffels@fz-juelich.de (G.S.); c.regio-brambilla@fz-juelich.de (C.R.B.); c.stegmayr@fz-juelich.de (C.S.); a.willuweit@fz-juelich.de (A.W.); w.worthoff@fz-juelich.de (W.A.W.); n.j.shah@fz-juelich.de (N.J.S.); c.lerche@fz-juelich.de (C.L.); p.lohmann@fz-juelich.de (P.L.); 2Department of Nuclear Medicine, RWTH Aachen University Hospital, 52074 Aachen, Germany; fmottaghy@ukaachen.de; 3Center of Integrated Oncology (CIO), Universities of Aachen, Bonn, Cologne and Duesseldorf, 53127 Bonn, Germany; 4Department of Neurology, Faculty of Medicine, University Hospital Cologne, University of Cologne, 50931 Cologne, Germany; 5Department of Stereotaxy and Functional Neurosurgery, Center for Neurosurgery, Faculty of Medicine, University Hospital Cologne, 50931 Cologne, Germany; martin.kocher@uk-koeln.de; 6Department of Neurology, RWTH Aachen University Hospital, 52074 Aachen, Germany; 7Department of Radiology and Nuclear Medicine, Maastricht University Medical Center (MUMC+), 6229 HX Maastricht, The Netherlands

**Keywords:** brain tumour diagnosis, cerebral glioma, PET, radiolabelled amino acids, O-(2-[^18^F]fluoroethyl)-L-tyrosine (FET), Hybrid PET/MRI, multimodal imaging

## Abstract

**Simple Summary:**

Advanced MRI methods and PET using radiolabelled amino acids provide valuable information in addition to conventional MR imaging for brain tumour diagnostics. The advent of hybrid PET/MRI has allowed a convergence of the methods, but up-to-date simultaneous imaging has reached little relevance in clinical neuro-oncology. A key factor for the benefit of PET/MRI in neuro-oncology is a multimodal approach that provides decisive improvements in the diagnostics of brain tumours compared with a single modality. This review focuses on studies investigating the additive value of amino acid PET and advanced MRI in the diagnosis of cerebral gliomas.

**Abstract:**

Advanced MRI methods and PET using radiolabelled amino acids provide valuable information, in addition to conventional MR imaging, for brain tumour diagnostics. These methods are particularly helpful in challenging situations such as the differentiation of malignant processes from benign lesions, the identification of non-enhancing glioma subregions, the differentiation of tumour progression from treatment-related changes, and the early assessment of responses to anticancer therapy. The debate over which of the methods is preferable in which situation is ongoing, and has been addressed in numerous studies. Currently, most radiology and nuclear medicine departments perform these examinations independently of each other, leading to multiple examinations for the patient. The advent of hybrid PET/MRI allowed a convergence of the methods, but to date simultaneous imaging has reached little relevance in clinical neuro-oncology. This is partly due to the limited availability of hybrid PET/MRI scanners, but is also due to the fact that PET is a second-line examination in brain tumours. PET is only required in equivocal situations, and the spatial co-registration of PET examinations of the brain to previous MRI is possible without disadvantage. A key factor for the benefit of PET/MRI in neuro-oncology is a multimodal approach that provides decisive improvements in the diagnostics of brain tumours compared with a single modality. This review focuses on studies investigating the diagnostic value of combined amino acid PET and ‘advanced’ MRI in patients with cerebral gliomas. Available studies suggest that the combination of amino acid PET and advanced MRI improves grading and the histomolecular characterisation of newly diagnosed tumours. Few data are available concerning the delineation of tumour extent. A clear additive diagnostic value of amino acid PET and advanced MRI can be achieved regarding the differentiation of tumour recurrence from treatment-related changes. Here, the PET-guided evaluation of advanced MR methods seems to be helpful. In summary, there is growing evidence that a multimodal approach can achieve decisive improvements in the diagnostics of cerebral gliomas, for which hybrid PET/MRI offers optimal conditions.

## 1. Introduction

Currently, the diagnosis of brain tumours is primarily based on contrast-enhanced MRI. Structural imaging using T1- and T2-weighted sequences provides high-resolution imaging of brain tumours and allows a differential diagnosis in a large fraction of lesions [1]. Differentiating tumour tissue from non-specific tissue changes, however, can be difficult, especially in cases of gliomas with diffusely infiltrating tumour growth, lack of contrast enhancement, and reactive tissue changes after surgery, radiotherapy, alkylating chemotherapy, or other experimental therapy approaches. In this situation, PET using radiolabelled amino acids can provide important additional diagnostic information [2]. The Response Assessment in Neuro-Oncology (RANO) Working Group has recommended the use of amino acid PET, in addition to MRI, in all stages of brain tumour management [3,4,5,6,7,8]. O-(2-[^18^F]-fluoroethyl)-L-tyrosine (^18^F-FET) was developed in our institution in the 1990s in order to provide a fluorine-18-labelled amino acid PET tracer with a longer half-life (110 min), which provides logistical advantages compared with shorter-lived carbon-11 labelled amino acids (half-life 20 min) such as [^11^C]-methyl-L-methionine [9,10,11]. Since 2000, we have focused on preclinical and clinical brain tumour imaging with ^18^F-FET, which has become one of the most frequently used amino acid tracers in the field [12,13]. Meanwhile, the interest of neuro-oncologists, neurosurgeons and radiation oncologists in ^18^F-FET PET has increased considerably, leading to 600–700 ^18^F-FET PET investigations per year in our department alone [12,14]. 

The introduction of PET/CT in the early 2000s constituted a milestone in nuclear medicine as it provided the precise anatomical localisation of abnormal tracer uptake in whole-body PET imaging. This has significantly improved diagnostic accuracy, and meanwhile PET/CT systems have replaced stand-alone PET scanners [15,16]. However, for brain imaging, the introduction of PET/CT was less important, because the rigid structure of the skull allows an efficient spatial co-registration of separately acquired PET, CT and MRI data [17]. 

Since around 2010, hybrid PET/MRI has become commercially available, representing another important development in the field. Although, like PET/CT, PET/MRI does not provide an essential advantage for the co-registration of images of brain tumour patients, the benefits relate more to an improved workflow, reduced examination time and, especially in paediatric patients, the avoidance of radiation exposure from the CT scanner and the repeated use of general anaesthesia [18]. Early reviews have highlighted the potential of simultaneous PET/MRI for the combination of various physiological parameters, MR-based motion, and partial volume correction and the optimised generation of arterial input function for metabolic modelling [19]. So far, however, these features have not had a major impact on clinical brain tumour diagnostics, and a recent paper has emphasised the equality of hybrid and sequential PET/MRI [20]. 

Our laboratory has been equipped with a dedicated BrainPET-hybrid PET/MRI system since 2008, in addition to an existing conventional PET system [21]. However, the hybrid scanner has only been used for approximately 25% of the ^18^F-FET PET investigations undertaken at our institute. In our experience, the more frequent use of hybrid PET/MRI is limited due to the fact that nearly all brain tumour patients have already received conventional MR imaging before referral for ^18^F-FET PET. Amino acid PET or advanced MRI are usually second-line investigations in patients with equivocal findings in conventional MRI (see flow chart in Figure 1). Most of the patients referred for ^18^F-FET PET have already had recent contrast-enhanced MRI scans, and a second injection of contrast medium for perfusion-weighted MRI (PWI) must be carefully weighed against clinical necessity. Moreover, our team perceives hybrid PET/MRI as more time-consuming than a PET or PET/CT scan due to checking for magnetic materials, sedation for claustrophobia or the refusal of additional MRI because of noise. Despite this, hybrid PET/MRI may be particularly useful when a second line examination with both amino acid PET and advanced MRI is intended and an additive diagnostic value can be expected.

Several reviews have discussed the technical aspects and the potential of hybrid PET/MRI in neuro-oncology, and it is not the intention of this review to repeat these aspects [18,22,23,24]. Instead, this review focuses on studies investigating the diagnostic value of combined amino acid PET and advanced MR methods in the diagnosis of brain tumours. The analysis is limited to cerebral gliomas, as no corresponding studies on other tumour entities, such as cerebral lymphomas or cerebral metastases, were found.

The following chapters first provide a short overview of PET and advanced MR methods in brain tumour diagnostics. Thereafter, we give a review of studies evaluating the additive or complementary value of these methods, providing a special perspective for the use of hybrid PET/MRI in neuro-oncological diagnostics.

## 2. Search Strategy

A PubMed search of the published literature with the combination of the search terms “glioblastoma”, “brain tumours”, “high-grade glioma”, “positron emission tomography”, “magnetic resonance imaging”, “magnetic resonance spectroscopy”, “perfusion-weighted imaging”, “diffusion-weighted imaging”, “chemical exchange saturation transfer”, “kurtosis”, “DKI”, “PET”, “amino acid PET”, “MRI”, “advanced MRI”, “MRS”, “PWI”, “DWI”, “CEST” and “hybrid PET/MR” before and inclusive of October 2022 was performed. Additional literature was retrieved from the reference lists of all identified articles. Furthermore, articles identified through searches of the authors’ files were included. Only publications in English were considered.

## 3. PET Tracers for Brain Tumour Imaging

Today, radiolabelled amino acids are the preferred PET tracers in neuro-oncology [1]. Amino acid PET is helpful regarding differential diagnosis, classification and the prognostication of newly diagnosed brain tumours, the delineation of brain tumour extent for treatment planning, the assessment of treatment response and the differentiation of tumour recurrence or progression from treatment-related changes [1]. The most widely used amino acid tracers are [^11^C]-methyl-L-methionine (MET), ^18^F-FET and 3,4-dihydroxy-6-[^18^F]-fluoro-L-phenylalanine (^18^F-FDOPA), as described in previous publications from the RANO Group [3,4]. Furthermore, the synthetic amino acid analogue anti-1-amino-3-[^18^F]fluorocyclobutane-1-carboxylic acid (FACBC or Fluciclovine) has gained clinical interest for brain tumour imaging in recent years [25,26,27]. The uptake of these tracers in brain tumours is primarily dependent on the increased expression and functionality of large neutral amino acid transporters of the L-type (LAT, subtypes LAT1 and LAT2) [1]. For more information on the various amino acid tracers, we refer the reader to a recently published review article [28]. In contrast to radiolabelled amino acids, the most widely used PET tracer 2-[^18^F]-fluorodeoxyglucose (^18^F-FDG) has a limited use in brain tumours because of the high glucose metabolism in normal brain tissue. The proliferation tracer [^18^F]-3′-deoxy-3′-fluorothymidine accumulates in cerebral gliomas in relation to the grade of malignancy and prognosis [29,30], but uptake is usually restricted to contrast-enhancing tumour parts on MRI and the tumour volume is smaller than that observed with amino acid tracers [31]. [^11^C]-choline or [^18^F]-fluoro-choline are markers of cell membrane phospholipids in brain tumours, but tracer uptake is also restricted to tumour parts with the disruption of the blood–brain barrier (BBB) [32]. A correlation of tracer uptake with the grade of malignancy has been reported [33,34], but the role of choline tracers in the primary diagnosis of brain tumours is limited, as the accumulation is not tumour-specific [35,36,37].

Many studies have explored brain tumour imaging with the hypoxia tracer [^18^F]-fluoromisonidazole (^18^F-FMISO) [38,39], and several review articles have summarised the present knowledge on this tracer [40,41,42]. There is widespread agreement that increased ^18^F-FMISO uptake correlates with tumour grade and prognosis [38,43], but the most challenging indication for ^18^F-FMISO PET, i.e., the effectiveness of radiotherapeutic dose escalation in hypoxic areas in gliomas, still remains unanswered [41,44].

Another important approach for brain tumour imaging is the use of ligands for the mitochondrial translocator protein (TSPO), such as [^11^C]-PK11195, [^18^F]-GE-180 and [^18^F]-DPA-714 [45]. TSPO is overexpressed in activated microglia and macrophages, but also in glioma cells [46]. PET imaging of gliomas using TSPO ligands depicts tumours with high contrast compared with the normal brain [47], but discrimination between tumour mass and brain tissue appears to be critical at the tumour rim, where glia-associated microglia/macrophages may also show high tracer binding [48,49,50]. TSPO ligands accumulate in brain areas with intact BBB, but differences exist in the visualisation of tumour extent compared with amino acid PET [51].

In addition to the tracers mentioned, a large number of other ligands are currently under development, and it is beyond the scope of this article to provide a complete overview. In this regard, reference is made to corresponding review articles [24,40,52]. Overall, none of those tracers has reached a clinical status comparable to that of radiolabelled amino acids. Therefore, this review focuses on the combination of amino acid PET and advanced MRI techniques.

## 4. Advanced MRI Methods in Neuro-Oncology

Advanced MRI methods can provide functional, physiologic and molecular information beyond conventional MRI, which may be helpful in equivocal findings [53]. A detailed description of these methods is beyond the scope of this article, and therefore only a brief overview of the most important methods from this area is given. PWI either via dynamic susceptibility contrast (DSC) MRI, dynamic contrast-enhanced (DCE) MRI or arterial spin labeling (ASL) MRI provides several surrogate markers of tissue perfusion, such as relative cerebral blood flow (rCBF), relative cerebral blood volume (rCBV), and other perfusion metrics [1,54,55]. In particular, rCBV mapping is a valuable supplement to conventional MRI in the differentiation of tumour progression or recurrence from treatment-related changes [56].

Proton MR spectroscopy (MRS) enables the non-invasive measurement of the signals of selected metabolites in vivo. Important metabolites for the characterisation of brain tumours are the neuronal marker N-acetyl-aspartate (NAA) and choline-containing compounds as cell membrane markers (Cho). MR spectroscopic imaging (MRSI) provides parameter maps, which visualise heterogenous distributions of different metabolites, or ratios thereof, in larger volumes of the brain [57]. Diffusion-weighted imaging (DWI) is based upon the random Brownian motion of water molecules within a voxel of tissue, which can be quantified, for example, by the apparent diffusion coefficient (ADC) [58]. In brain tumours, the ADC is inversely correlated with cell density, probably due to reduced water mobility from dense cellular packing. Diffusion kurtosis imaging (DKI) is an advanced neuroimaging modality that is an extension of diffusion tensor imaging by estimating the kurtosis (skewed distribution) of water diffusion based on a probability distribution function [59]. Another approach uses a combination of magnetisation transfer contrast and spectroscopic techniques based on the chemical exchange saturation transfer (CEST) effect [60,61]. The CEST effect from amides allows the imaging of amide proton transfer (APT), which appears to be related to the tumour extent of cerebral gliomas.

Another promising field for the investigation of brain tumours is sodium imaging via single-quantum and multiple-quantum ^23^Na MRI and spectroscopy [62]. Cell membrane depolarisation that precedes cell division in proliferative neoplastic tissue leads to an increase in the intracellular sodium concentration and a concomitant rise in the total sodium concentration in the tumour tissue [63]. Initial investigations have addressed the treatment monitoring and analysis of the IDH mutation status of gliomas [64,65].

## 5. Hybrid PET/MRI in Animal Research

Hybrid PET/MRI has been successfully used in preclinical neuroimaging to correlate changes in neuronal activity using fMRI and changes in receptor expression and neurotransmitter binding [66,67,68,69]. Simultaneous PET/MRI imaging is essential for these examinations, as neuronal activations in temporally separate examinations are not comparable and do not permit any reliable conclusions. Moreover, several studies have used combined PET and MRI in animal brain tumour models to explore novel PET tracers and advanced MR methods for brain tumour diagnosis, but the investigations have used mainly sequential PET/MRI [70,71,72,73,74].

Previous review articles have made suggestions as to the expectation that simultaneous hybrid PET/MRI will be used for the modelling of physiological and biochemical processes, because during the simultaneous acquisition one can be sure the prevailing physiological conditions such as blood flow and perfusion, pertain to both the PET and MRI measurements [75]. However, there has been little implementation in experimental brain tumour research to date. Nevertheless, hybrid PET/MRI offers decisive logistical advantages in animal imaging, as the standard sequential execution of PET and MRI considerably prolongs examination times or leads to examinations on different days, requiring renewed vascular puncture and anaesthesia. Thus, hybrid PET/MRI provides considerable advantages in terms of animal welfare and reducing the number of animal experiments. Due to the lack of an animal hybrid PET/MRI scanner in our department, we have successfully worked with a fixed animal bed, which allows rapid sequential PET/MRI without re-anaesthesia [75,76].

## 6. Hybrid PET/MRI in Newly Diagnosed Cerebral Gliomas

In brain lesions suspicious for neoplasms, conventional MRI is frequently inconclusive and additional imaging methods can be helpful. This concerns differential diagnosis, the definition of an optimal biopsy site, and the detection of tumour infiltration, especially in tumours without contrast enhancement in MRI. Furthermore, the non-invasive classification of tumours and the assessment of molecular features and prognostication can be valuable if neuropathological assessment is not possible. Pyka et al. investigated the value of combined ^18^F-FET PET and MRS in a series of 67 patients with newly diagnosed gliomas [77]. Static ^18^F-FET PET allowed the differentiation of low-grade and high-grade gliomas with an area under the curve (AUC) in receiver operating characteristics analysis (ROC) of 0.86 and MRS using the Cho/NAA with an AUC of 0.72. The combination of ^18^F-FET PET and MRS achieved an AUC of 0.97. Furthermore, the multimodal approach was able to differentiate glioblastoma from non-glioblastoma with an AUC of 0.97. In the survival analysis, PET parameters (but not spectroscopy) were significantly correlated with overall survival. Song et al. reported that the combination of ^18^F-FET PET and DSC-PWI increased the diagnostic accuracy to differentiate gliomas with and without IDH mutation (AUC 0.90) compared with the single modalities (^18^F-FET PET and rCBV, each AUC 0.80), but none of the parameters discriminated between oligodendrogliomas and astrocytomas [78]. Haubold et al. explored the non-invasive characterisation of cerebral gliomas utilising multi-parametric ^18^F-FET PET/MRI and MR fingerprinting in a series of 42 patients with suspected primary brain tumour [79]. For the differentiation of low-grade and high-grade gliomas, the combination with ^18^F-FET PET yielded the highest AUC value (0.85), but most parameters (i.e., 1p19q co-deletion, ATRX, IDH-status, MGMT promotor mehtylation, WHO subtype) could be best estimated with MR parameters alone. The potential of amino acid PET for the assessment of the tumour extent of gliomas has been documented by several biopsy-controlled studies [80,81,82,83,84,85]. Most studies have compared the tumour extent in amino acid PET with conventional MRI, but initial studies also considered advanced MRI methods for comparison [86,87,88]. Several studies have compared tumour extent in amino acid PET with rCBV mapping and have demonstrated significant differences between the methods [89,90,91,92]. Therefore, there is widespread consensus that rCBV imaging is not suitable for the tumour delineation of gliomas. In another prospective, biopsy-controlled study, the detection of tumour extent using ^18^F-FET PET was compared with different advanced MR methods [87]. One hundred and seventy-four tissue samples were taken from 20 patients, and the contribution of ^18^F-FET PET, PWI, DWI, APT-CEST and MRSI to delineate the tumour tissue was analysed with multiple logistic regression. It was found that the combination of ^18^F-FET PET and ADC mapping best reflected the tumour extent. The contribution of MRSI could not be evaluated due to multiple artifacts in this series of patients. Another study compared tumour spread with the ^18^F-FET PET, APT-CEST, and PWI of newly diagnosed gliomas [88]. The tumour extent seemed to be comparable with both APT CEST and ^18^F-FET PET and correlated well with cell density. In a study using ultra-high field MRI at 7T, APT CEST predicted the tumour extent using ^18^F-FET PET as a reference with an AUC of 0.81 and MRS with an AUC of 0.89 [93]. The combination of APT-CEST and MRS predicted ^18^F-FET uptake with an AUC of 0.95. The authors concluded that the combination of APT-CEST and MRS might serve as an alternative to amino acid PET to delineate glioma infiltration. An overview of studies demonstrating an additive value of amino acid PET and advanced MR methods in newly diagnosed cerebral gliomas is given in Table 1.

Summarising, there is some evidence that combined amino acid PET and advanced MRI is helpful in improving the non-invasive characterisation of suspected gliomas. Concerning tumour delineation, amino acid PET appears to be the most reliable method to identify metabolically active tumour tissue, and so far there is little evidence that the combination with advanced MR methods leads to superior results.

## 7. Hybrid PET/MRI in Patients with Recurrent Gliomas

Most studies investigating multimodal PET/MRI to differentiate brain tumour progression or recurrence from treatment-related changes have compared PWI with amino acid PET. While some older publications reported the superiority or equivalence of rCBV mapping compared with amino acid PET [94,95,96], more recent publications consistently observed the superiority of amino acid PET [97,98,99]. Recently, we analysed the additive value of ^18^F-FET PET and perfusion-weighted MRI in a group of 104 patients with suspected glioma recurrence [100]. Eighty-three patients had tumour progression and 21 patients had treatment-related changes. The combination of ^18^F-FET PET and PWI did not increase the diagnostic power, but an rCBVmax > 2.85 reached a positive predictive value of 100% so that 44 patients could be correctly classified using rCBVmax alone. In the remaining patients, ^18^F-FET PET still achieved an accuracy of 78%, so that 87% of the patients could be correctly diagnosed, in total. These results support the sequential use of PWI and amino acid PET, particularly when a more economical use of the diagnostic methods has priority. In contrast, one study using ^11^C-MET PET reported on an additive value of amino acid PET and DSC-PWI [101]. While both the maximum tumour-to-brain ratio (TBR_max_) of ^11^C-MET uptake and mean rCBV achieved an AUC of 0.85, the combination of the parameters yielded an AUC of 0.95 in the differentiation tumour recurrence from radiation injury. Furthermore, a number of studies have reported the additive value of amino acid PET and MRI when including advanced MRI methods other than rCBV in patients with suspected tumour recurrence. Jena et al. achieved the highest accuracy (97%) in differentiating recurrent tumours from radiation necrosis when combining the TBR_max_ of ^18^F-FET uptake and MRS using the Cho/Cr ratio [102]. An identical accuracy of 97% was achieved by Sogani et al. with a combination of ^18^F-FET PET, MRS, PWI and DWI [103], and a hybrid PET/MRI study achieved an accuracy of 95% using ^18^F-FDOPA as the amino acid tracer [104]. Another hybrid PET/MRI study compared dynamic ^18^F-FET PET, PWI, and DWI in 47 patients with suspected glioma recurrence [105]. Static ^18^F-FET PET alone achieved an AUC of 0.86 for differentiating recurrent tumour and treatment-related changes, which could be increased to an AUC of 0.89 when combined with PWI and DWI. Lohmeier et al. reported the highest AUC by using a combination of static ^18^F-FET PET and ADC (0.90) versus ^18^F-FET PET (0.81) or ADC alone (0.82) [106]. These results could not be confirmed by Werner et al., who reported the highest accuracy using static and dynamic ^18^F-FET PET parameters (93%), which could not be further improved by ADC mapping [107].

A recent study applied a machine learning approach to a multiparametric data set of 66 patients with suspected tumour recurrence, including ^18^F-FET PET, DSC-PWI and APT-CEST [108]. The classification accuracy of the Random Forest classifier was 0.86 and therefore significantly above the no-information rate of 0.77 compared to an accuracy of 0.82 for MRI, 0.81 for ^18^F-FET PET, and 0.81 for expert consensus. These results emphasise that the use of artificial intelligence in conjunction with multiparametric imaging can be expected to yield further improvements in diagnostic accuracy. Rather encouraging results could be observed by our group with the combination of ^18^F-FET PET and DKI in patients with recurrent glioma [109]. In this study, the ^18^F-FET PET-guided evaluation of kurtosis achieved an AUC of 0.87 (MK-C90), ^18^F-FET uptake an AUC of 0.77 (TBR_max_), and the combination of the two methods achieved an AUC of 0.97 to differentiate recurrent tumours from treatment-related changes (Figure 2 and Figure 3). These data were confirmed by a recent study including 87 patients with suspected recurrent glioblastoma using ^11^C-MET [110]. In that study, combined ^11^C-MET PET and DKI achieved an AUC of 0.95 to differentiate glioblastoma recurrence from radiation injury compared with an AUC of 0.89 for PET or 0.85 for DKI alone.

Few data exist concerning the additive value of amino acid PET and advanced MR methods in terms of response assessment. A recent study reported that the simultaneous evaluation of ^18^F-FET PET and ADC metrics using PET/MRI allowed the early and reliable identification of treatment responses and predicted overall survival in recurrent glioblastoma patients treated with regorafenib [111]. A key aspect in this study was the fact that the authors used pathological ^18^F-FET uptake to define the region of interest (ROI) on the ADC maps. The authors emphasised that radiological recommendations do not provide a strategy for identifying the ROI on the DWI-ADC images or how to define the threshold for pathological ADC values. Thus, a PET-guided evaluation strategy for advanced MRI methods is another important aspect for the use of PET/MRI and also played a decisive role in the combined use of ^18^F-FET PET and DKI mentioned above [109]. An overview of studies demonstrating an additive value of amino acid PET and advanced MR methods in recurrent cerebral gliomas is given in Table 2.

## 8. Hybrid PET/MRI in Paediatric Brain Tumours

The use of hybrid PET/MRI appears particularly advantageous in paediatric patients, in order to reduce the examination time, to avoid radiation exposure from the CT scanner, and to prevent repeated general anaesthesia in separate measurements [112,113]. Furthermore, the fusion of separately acquired PET and MRI data may cause more problems in children than in adults owing to the fact that paediatric tumours are frequently located in the cerebellum and medulla or by high extra cerebral ^18^F-FET uptake in the cranial bone marrow [18]. On the other hand, the logistics of anaesthesia in the hybrid scanner are challenging, especially in younger children, and attenuation correction in children causes problems [18] as MR-based attenuation methods often are built upon reference data sets acquired in adult subjects [114,115]. Several studies have demonstrated the additional value of amino acid PET in paediatric brain tumours compared with conventional MRI [7,116,117,118]. It was reported that amino acid PET changed patient management in up to two-thirds of children and adolescents with brain tumours [113,119]. The first data on the complementary value of amino acid PET and advanced MRI methods in paediatric brain tumours are available. In a comparative study between ^18^F-FDOPA PET and ^1^H-MRS in 27 children with untreated brain tumours, PET was superior in tumour grading and prognostication while ^1^H-MRS was better in differentiating tumours from non-neoplastic lesions [117]. Another study in 26 children with diffuse astrocytic gliomas yielded the highest diagnostic performance in predicting tumour progression when combining ^18^F–DOPA PET, ADC, and arterial spin labelling data [120]. Thus, there is initial evidence of an additional value of amino acid PET and advanced MRI methods in the assessment of childhood brain tumours.

## 9. Conclusions

In principle, all applications of combined amino acid PET and advanced MRI in brain tumours mentioned in this review do not require simultaneous acquisition and can be performed sequentially. Hybrid PET/MRI is preferable to reduce examination time and, particularly in children, to reduce radiation burden and repeated anaesthesia. There is increasing evidence that the combination of amino acid PET and advanced MRI improves grading and molecular characterisation in newly diagnosed tumours, while data concerning the delineation of tumour extent and biopsy guidance are limited. Convincing and clinically relevant additive diagnostic value is achieved by combining amino acid PET with different advanced MR methods regarding the differentiation of tumour progression or recurrence versus treatment-related changes. In this context, the value of the PET-guided evaluation of advanced MR methods should be emphasised, as defining the region of interest in these methods can be difficult.

## Figures and Tables

**Figure 1 cancers-15-03577-f001:**
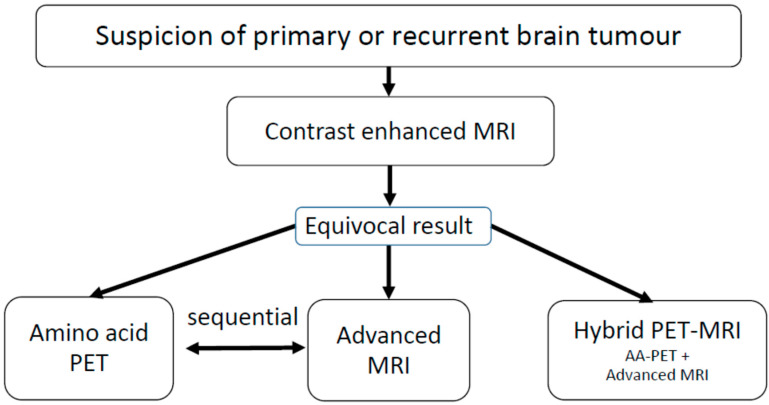
Workflow in brain tumour imaging: When there is suspicion of a primary or recurrent brain tumour, the first step is conventional, contrast-enhanced MRI. If the findings are equivocal, further diagnostics using amino acid PET or advanced MRI procedures are considered. At this point, hybrid PET/MRI may be advantageous if a combination of these methods can achieve higher accuracy compared with a single modality.

**Figure 2 cancers-15-03577-f002:**
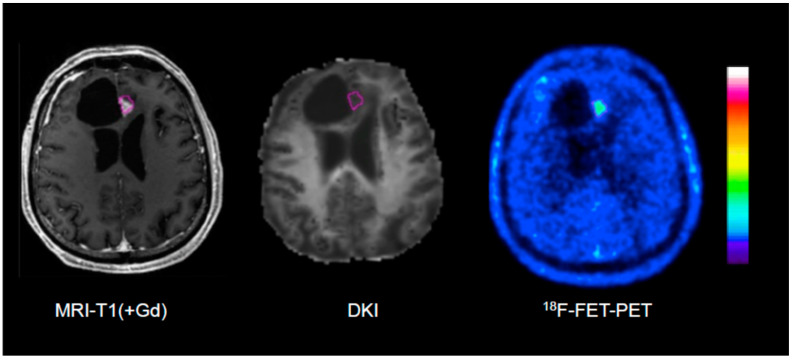
Example of an ^18^F-FET PET-guided evaluation of diffusion kurtosis imaging (DKI) in a patient with treatment-related changes. Please note that the region of interest (pink line) generated on the PET scan (**right**) is larger than the area of contrast enhancement in T1-weighted MRI (**left**).

**Figure 3 cancers-15-03577-f003:**
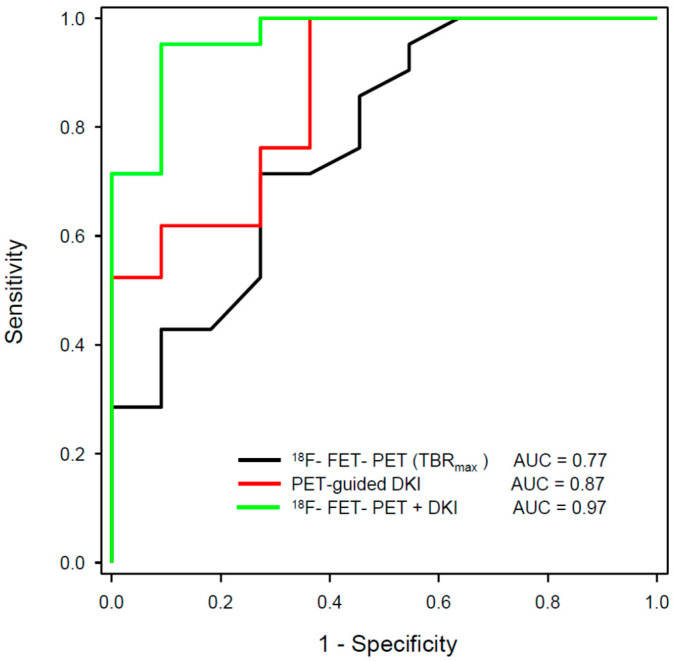
ROC analysis for differentiation between the tumour progression and treatment-related changes in gliomas using hybrid PET/MRI with ^18^F-FET PET and diffusion kurtosis imaging (DKI) from a previous publication of our group [109]. The largest area under the curve (AUC) could be achieved by the combination of ^18^F-FET PET and ^18^F-FET PET-guided DKI (green line).

**Table 1 cancers-15-03577-t001:** Studies demonstrating an additive value of amino acid PET and advanced MR methods in newly diagnosed cerebral gliomas.

Reference	Year	PET Tracer	MR-Methods	Tumour Type	No of Subjects	Remarks	Main Result	Cutoff Values
Verburg et al. [87]	2020	^18^F-FET	PWI, DWI, MRS	Newly diagnosed gliomas	20	Tumour infiltration, Verification of tumour extent by biopsies	Best result for combined ^18^F-FET + ADC in depicting enhancing gliomas	n.a.
Haubold et al. [79]	2020	^18^F-FET	DWI, ADC, SWI	Phenotyping of newly diagnosed gliomas	42	Radiomics, multiparametric MRI and ^18^F-FET PET parameters	Best differentiation of high-grade and low-grade glioma by combination of ^18^F-FET PET, T1ce and SWI	n.a.
Song et al. [78]	2021	^18^F-FET	PWI	Phenotyping of newly diagnosed gliomas	52	Retrospective evaluation after surgery	Improved differentiation of IDH status by combination of ^18^F-FET PET and PWI.	^18^F-FET TBRmean > 1.91, ^18^F-FET TBRmax > 3.81, nCBVmean > 1.04
Pyka et al. [77]	2022	^18^F-FET	MRSI	Newly diagnosed gliomas	67	Characterisation of intracranial gliomas	Improved differentiation of high-grade from low-grade glioma and of glioblastoma from non-glioblastoma.	^18^F-FET TBRmean > 2.00, ^18^F-FET Time-to-peak < 20 min, MRS NAA/Cr > 1.89, MRS Cho/Cr > 2.22,

**Table 2 cancers-15-03577-t002:** Studies demonstrating additive value of amino acid PET and advanced MR methods in recurrent brain tumours.

Reference	Year	PET Tracer	MR Methods	Tumour Type	No of Subjects	Remarks	Main Result	Cutoff Values
Jena et al. [102]	2016	^18^F-FET	PWI, DWI, MRSI	Tumour recurrence in pretreated gliomas	26	Verification by surgery (9) and clinical follow-up (17)	Best AUC by combination of ^18^F-FET PET, rCBV and MRS (0.94) versus ^18^F-FET PET (0.89), ADC (0.74), PWI (0.85), MRS (0.89)	^18^F-FET TBRmean > 1.44, ^18^F-FET TBRmax > 2.11, rCBVmean > 1.89, ADCmean < 1611, MRS Cho/Cr > 1.42
Sogani et al. [103]	2017	^18^F-FET	PWI, DWI, MRSI	Tumour recurrence in pretreated gliomas	32	Verification by surgery (12) and clinical follow-up (20)	Best accuracy by combination of ^18^F-FET PET, ADC, rCBV and MRS (97%)	^18^F-FET TBRmean > 1.52, ^18^F-FET TBRmax > 2.09, rCBVmean 1.78, ADCmean 1594, MRS Cho/Cr 1.54
Pyka et al. [105]	2018	^18^F-FET	PWI, DWI	Tumour recurrence in pretreated gliomas	47 (63 lesions)	Verification by surgery (23) and clinical follow-up (40)	Improved accuracy by combination of ^18^F-FET PET, ADC and rCBV (AUC 0.89)	^18^F-FET TBRmean > 2.07, ^18^F-FET Time-to-peak < 20 min, rCBVmean corr. > 3.35, ADCmean < 1610
Lohmeier et al. [106]	2019	^18^F-FET	DWI-ADC	Recurrent high- and low-grade gliomas	42	Verification by surgery (36) and clinical follow-up (6)	Best AUC by combination of static ^18^F-FET PET and ADC (90%) versus ^18^F-FET PET (0.81) or ADC alone (0.82)	^18^F-FET TBRmax > 2.0, ADCmean < 1254
Qiao et al. [101]	2019	^11^C-MET	PWI-DSC	Recurrent high- and low-grade gliomas	42	Verification by surgery (32) and clinical follow-up (10)	Best AUC by combination of ^11^C-MET PET and rCBV (0.95) versus ^11^C-MET PET (0.85) or rCBV alone (0.85)	^18^F-FET TBRmax > 1.85, rCBVmean > 1.83,
Paprottka et al. [108]	2021	^18^F-FET	APT-CEST, PWI	Tumour recurrence in pretreated gliomas	66 (74 lesions)	Verification by surgery (46) and clinical follow-up (31), ADC evaluation guided by ^18^F-FET PET	Best accuracy by combination of ^18^F-FET PET, APT-CEST and PWI (0.85) versus ^18^F-FET PET alone (0.81)	n.a.
D’Amore et al. [109]	2021	^18^F-FET	DWI, DKI	Tumour recurrence in pretreated gliomas	32	Verification by surgery (12) and clinical follow-up (20), DKI evaluation guided by ^18^F-FET PET	Best AUC by combination of static ^18^F-FET PET and DKI (0.97) versus ^18^F-FET PET (0.77) or DKI alone (0.87)	^18^F-FET TBRmax > 2.95, MK C90 > 0.62, ^18^F-FET-DKI index > 41
Jena et al. [104]	2021	^18^F-FDOPA	PWI, DWI, MRS	Tumour recurrence in pretreated gliomas	26	Verification by surgery (4) and clinical follow-up (22)	Best AUC by combination of ^18^F-FDOPA PET, rCBV, ADC and MRS (0.94) versus ^18^F-FDOPA-PET (0.81), ADC (0.42), rCBV (0.50) and MRS (0.77) alone	n.a.
Lombardi et al. [111]	2021	^18^F-FET	DWI	Monitoring of regorafenib therapy in recurrent glioblastoma	16	Verification by clinical follow-up, ADC evaluation guided by ^18^F-FET PET	^18^F-FET guided ADC promising for therapy monitoring, better than RANO	n.a.
Dang et al. [110]	2022	^11^C-MET	DWI, DKI	Tumour recurrence in pretreated gliomas	86	Verification by surgery (23) and clinical follow-up (20)	Best AUC by combination of ^11^C-MET PET and DKI (0.95).	^18^F-FET TBRmax > 2.13, MK > 0.81, combined PET/MRI model > 0.17

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
