# Peer review of "Hybrid PET/MRI in Cerebral Glioma: Current Status and Perspectives"

_cancers, 2023, doi:10.3390/cancers15143577_

Round 1
Reviewer 1 Report
This is a brief well referenced review of role of amino-acid PET and sophisticated MRI in neuro-oncology. It provides a good resource for appraisal of these modalities and the authors provide experienced interpretation of the data sources. There are no corrections suggested to these data review, and believe it is suitable for publication in the themed journal.
The only concern with the manuscript is that the focus is titled to be based on Hybrid PET-MRI but given limited data it combines the data from sequential imaging rather than hybrid imaging research. However the authors do make frequent comments back to Hybrid imaging when available. Additionally they are honest in regards to the limited advantages of the modality and where there may be a niche for research or clinical application.
Author Response
Reply: We thank the reviewer for the positive comments. Indeed, we have also discussed some studies with sequential PET/MRI data, insofar as these investigated the combined value of PET and MRI and thus support the hypothesis of a benefit of the hybrid technique as outlined in the introduction. Therefore, we would like to retain the word ‘hybrid’ in the title as it is.
Reviewer 2 Report
I thank the authors for the opportunity to review the result of their work.
This article underlines the role of hybrid PET/MRI in neuro-oncology.
All clinicians are all too well aware of how complex diagnosis is in neuro-oncology.
The authors, while not doing a meta-analysis, have well defined the rationale of using a hybrid PET/MRI in the diagnosis of primary tumors and relapses.
Fig. 1 well represents clinician daily routine: hybrid MRI/PET would optimize the timing and reduce patients and care-givers anxiety.
I would stress, however, that the use of this radiological tecnique is currently limited in research centres.
I appreciated the work even if the optimal would have been a systematic litterature review or a meta-analysis also of pre-clinical data.
I have no particular comments.
Author Response
Reply: We thank the reviewer for the valuable comments. To our knowledge, hybrid PET/MRI is also available in university hospitals and large clinical centers in Europe and the United States and is thus not only available in research centers. As outlined in the introduction, this paper represents a systematic literature review of papers investigating the combined value of amino acid PET and advanced MR methods. Papers that only compared the two methods without investigating the value of combined imaging were ignored. In this respect, we have referred to the corresponding literature at various points in the manuscript
Reviewer 3 Report
It is a nice review about the role of hybrid amino acid PETMR in glioma.
1. (Title). Your manuscript only covers the " amino acid PET" and " glioma". I suggest modifying the title of your paper to " Hybrid amino acid PET/MRI in Glioma: Current Status and Perspectives". You didn't include other neuro-oncologic conditions, such as metastasis and CNS lymphoma. However, if you prefer to keep this title, please consider adding a discussion about metastasis and CNS lymphoma as well.
2. (Introduction): Please consider adding a new paragraph about the uptake, transportation, and metabolism of the amino acid tracers discussed in this paper.
3. Page 3: "Instead, this review focuses on our own studies comparing 18F-FET PET with advanced MR methods and studies from the literature that were able to demonstrate an additive value of amino acid PET and advanced MRI in the diagnosis of brain tumors. " This part is a bit unclear. Does it mean you have reviewed the published papers in this field and included papers that show the added value of PETMR, and you ignored papers that showed PETMR is not superior to MRI or PET alone? If so, you present only the promising papers, and you are overestimating the performance of PETMR. Please clarify this part.
4. Page 5. PETMR is now being used in daily clinical practice. The paragraphs about the application of PETMR in the animal is irrelevant, and I suggest deleting this part.
5. Page 6. "One study compared preoperative imaging with 11C-MET PET and PWI in oligodendrogliomas with histological sections after en bloc resection of the tumors [85]. 11C-MET accumulation correlated well with cell density and reliably reflected the extent of the tumor tissue, while CBV mapping did not correlate with neuropathological markers for tumor cells such as IDH1-mutated protein and Ki67 (proliferating cells), which were used to delineate the tumor. That study confirmed the observation of other studies that rCBV is not suitable for tumor delineation". .. It is known that oligodendroglioma is an exception in neuroimaging, and the MR perfusion parameters don't match with tumor grade and infiltration. I suggest deleting this part
6. Tables 1 and 2. For each paper, please add the cutoff value of PET and MR parameters for the reported AUC (e.g. SUV, rCBV,....) so the readers can understand which SUV or rCBV can differentiate different conditions ( high grade versus low grade, recurrence versus radiation necrosis).
7. Figure 2. There is a typo: " note that note that"
Author Response
Comments and Suggestions for Authors
It is a nice review about the role of hybrid amino acid PETMR in glioma.
- (Title). Your manuscript only covers the " amino acid PET" and " glioma". I suggest modifying the title of your paper to " Hybrid amino acid PET/MRI in Glioma: Current Status and Perspectives". You didn't include other neuro-oncologic conditions, such as metastasis and CNS lymphoma. However, if you prefer to keep this title, please consider adding a discussion about metastasis and CNS lymphoma as well.
Reply: We thank the reviewer for the valuable comments. Indeed, the analysis is limited to cerebral gliomas, as no corresponding studies on other tumor entities such as cerebral lymphomas or cerebral metastases, were found. We have added a corresponding sentence at the end of the introduction. We have changed the title as suggested.
- (Introduction): Please consider adding a new paragraph about the uptake, transportation, and metabolism of the amino acid tracers discussed in this paper.
Reply: We think that a more detailed description of the transport mechanisms of the different amino acid tracers is beyond the scope of this review. We have referred the readers to a recently published review article by our working group.
- Page 3: "Instead, this review focuses on our own studies comparing 18F-FET PET with advanced MR methods and studies from the literature that were able to demonstrate an additive value of amino acid PET and advanced MRI in the diagnosis of brain tumors. " This part is a bit unclear. Does it mean you have reviewed the published papers in this field and included papers that show the added value of PETMR, and you ignored papers that showed PETMR is not superior to MRI or PET alone? If so, you present only the promising papers, and you are overestimating the performance of PETMR. Please clarify this part.
Ad 3: Thank you for pointing out this possible misunderstanding. Of course, this systematic review includes all studies that have investigated the value of combined amino acid PET and advanced MR methods. The results of studies that did not find a benefit of the combination are referred to in various places in the manuscript. We have changed and modified the text in various places to avoid this misunderstanding. This paper is definitively not overestimating the performance of PET/MRI because it very critically describes the limited clinical application of the technique and critically mentions previous reviews highlighting the potential, which are not relevant for clinical practice.
- Page 5. PETMR is now being used in daily clinical practice. The paragraphs about the application of PETMR in the animal is irrelevant, and I suggest deleting this part.
Reply: The importance of hybrid PET/MRI in animal studies is independent of its application in humans, as this method can be used to investigate various scientific questions that require the simultaneous application of PET and MRI, such as the mentioned studies on neuronal activity and receptor expression. Reviewer 2 even calls for a more systematic meta-analysis of the preclinical data. We have added an explanatory sentence and would like to keep the paragraph about animals in its present form.
- Page 6. "One study compared preoperative imaging with 11C-MET PET and PWI in oligodendrogliomas with histological sections after en bloc resection of the tumors [85]. 11C-MET accumulation correlated well with cell density and reliably reflected the extent of the tumor tissue, while CBV mapping did not correlate with neuropathological markers for tumor cells such as IDH1-mutated protein and Ki67 (proliferating cells), which were used to delineate the tumor. That study confirmed the observation of other studies that rCBV is not suitable for tumor delineation". It is known that oligodendroglioma is an exception in neuroimaging, and the MR perfusion parameters don't match with tumor grade and infiltration. I suggest deleting this part.
Reply: We have removed the section describing the results in oligodendroglioma and have focused on other studies demonstrating the discrepancy of amino acid PET and PWI for tumor delineation.
- Tables 1 and 2. For each paper, please add the cutoff value of PET and MR parameters for the reported AUC (e.g. SUV, rCBV,....) so the readers can understand which SUV or rCBV can differentiate different conditions ( high grade versus low grade, recurrence versus radiation necrosis).
Reply: We have inserted the cut-off values as far as they were available in the publications.
- Figure 2. There is a typo: " note that note that"
Reply: The typo has been corrected.
Round 2
Reviewer 3 Report
In "short summary" and "abstract" you used the term " systematic review". Your paper is not a systematic review. Please change it to " review" alone.